# LEARNING TO GENERALIZE COMPOSITIONALLY BY TRANSFERRING ACROSS SEMANTIC PARSING TASKS

## ABSTRACT

Neural network models often generalize poorly to mismatched domains or distributions. In NLP, this issue arises in particular when models are expected to generalize compositionally, that is, to novel combinations of familiar words and constructions. We investigate learning representations that facilitate transfer learning from one compositional task to another: the representation and the task-specific layers of the models are strategically trained differently on a pre-finetuning task such that they generalize well on mismatched splits that require compositionality. We apply this method to semantic parsing, using three very different datasets, COGS, GeoQuery and SCAN, used alternately as the pre-finetuning and target task. Our method significantly improves compositional generalization over baselines on the test set of the target task, which is held out during fine-tuning. Ablation studies characterize the utility of the major steps in the proposed algorithm and support our hypothesis.

## 1 INTRODUCTION

Recent work has spotlighted significant shortcomings of neural network approaches to NLP in coping with compositional generalization (CG) (Lake & Baroni, 2018; Finegan-Dollak et al., 2018; Keysers et al., 2020; Kim & Linzen, 2020; Shaw et al., 2021). In those studies, the test set combines in unfamiliar ways linguistic elements that may themselves be familiar from the training set; for example, sentences in the test set may be longer than those observed in training, or may use familiar words in syntactic roles in which they did not occur in training. The performance of popular neural architectures in contemporary NLP models such as Transformers drops considerably when they are expected to generalize compositionally. Several approaches have been proposed to address this issue, including specialized architectures with compositional inductive biases (Furrer et al. 2020) and compositional data augmentation (Andreas, 2020), but as of yet the problem is far from solved.

In this paper, we study this challenge from the perspective of transferability of compositional generalization skills: is possible for a neural model to transfer compositional generalization skills acquired from one task to another task that also requires compositional generalization skills?

We ground our inquiries with the semantic parsing tasks on three very different datasets: GeoQuery (Zelle, 1995), COGS (Kim & Linzen, 2020) and SCAN (Lake & Baroni, 2018). For each task, we use existing compositional training/test splits or create new ones. We propose a learning algorithm that can extract a compositional inductive bias from one task — a stage we refer to as pre-finetuning[1] — and transfers that bias to a target task, improving the models' compositional generalization behavior on that task.

To extract the inductive bias so as to transfer, we introduce a new training algorithm DUEL. In summary (cf. Fig 1), DUEL is designed to be compatible with pre-trained neural encoder-decoder models, such as T5 (Raffel et al., 2020). We view the encoder as learning a representation for the inputs and the decoder as "a task head" that is specialized to different tasks. In pre-finetuing, framing the task's two (compositional) splits as deriving representation from one split for zero-shot learning to the other split, DUEL trains the encoder and the decoder using different splits. In contrast to

---

[1]We borrow this terminology from the NLP research community. A large number of research papers have explored the idea of transfer learning, starting from a pre-trained language model, followed by training on a set of pre-finetuning tasks, and then fine-tuning on the target or downstream tasks (Vu et al., 2020; Gururangan et al., 2020; Pruksachatkun et al., 2020; Chen et al., 2020a; Aghajanyan et al., 2021).

using standard supervised learning over both splits as a whole for pre-finetuning, DUEL encourages the encoder to learn to represent the input texts in a way that facilitates greater compositional generalization, and that this transfers across domains and persists through fine-tuning on the target task, as shown by our empirical studies.

The rest of the paper is organized as follows. In §2, we detail our setup. We describe our approach in §4, followed by empirical studies §5. We discuss related work in §6 and conclude in §7.

## 2  PROBLEM SETUP

Our neural learner is given a pre-finetuning task and a target task. For the pre-finetuning task, we have a training data distribution $s$ and a test/eval data distribution $\tilde{s}$ that may deviate from $s$. Likewise, for the target task, we have $q$ and $\tilde{q}$. In the case of compositional generalization, the difference between $s$ and $\tilde{s}$ (or $q$ versus $\tilde{q}$) is defined in compositional terms: the ways in which elements combine differ systematically between the two parts of each datasets (Keysers et al., 2020; Kim & Linzen, 2020; Lake & Baroni, 2018; Hupkes et al., 2020).

Our goal is to train a neural model on ($s$, $\tilde{s}$) and then fine-tune it on $q$ so that the final model performs well on $\tilde{q}$, namely, attaining strong compositional generalization performance on the target task.

Our assumption is that the difference between $s$ and $\tilde{s}$ is similar to the difference between $q$ and $\tilde{q}$, such that an inductive bias acquired from ($s$, $\tilde{s}$) can be transferred to the target task. A motivating example is that $\tilde{s}$ contains longer texts than $s$ and so does $\tilde{q}$ than $q$: in this case, we would investigate whether learning to generalize from short to long texts is a transferrable skill.

## 3  TASK, DATA SETS AND THEIR COMPOSITIONAL SPLITS

**Data Splits**   In this work, we focus on semantic parsing, the task of translating a natural language text into a formal representation of its semantics. Standard data splits for supervised learning partition a dataset randomly into a training set and testing set. As such, the examples in both portions are drawn from the same distribution. By contrast, compositional splits partition a dataset in a way that creates a distributional discrepancy between the training and the test set, such that strong performance on the test set requires compositional generalization from the training set.

Compositional splits are straightforward to create for datasets that are generated from grammars or templates, such as SCAN (Lake & Baroni, 2018) and COGS (Kim & Linzen, 2020), described in detail in the next section. For instance, the templates used for the test set can include specific words in syntactic roles they did not appear in in the training set, or they can nest syntactic constituents more deeply than in the training set, creating differences in lengths of the input texts.

An alternative mechanism creates—based on an existing dataset for which the generative process is not available (for example, real-world user queries)—test sets that require compositional generalization by on splitting the dataset so as to obtain the *maximum compound divergence* (MCD) between the training and test examples (Keysers et al., 2020). Here, compounds are complex expressions formed by composing atoms, and compound divergence is computed as

$$\mathcal{D}_C(V\|W) = 1 - C_{0.1}(\mathcal{F}_C(V)\,\|\,\mathcal{F}_C(W))$$

where $\mathcal{F}$ is the distribution over compounds, and $C_\alpha(P\|Q) = \sum_k p_k^\alpha\, q_k^{1-\alpha}$. The intuition behind this method is that given an existing dataset it is difficult to limit the test examples to compounds that are completely *absent* from the training set, but by making the compound distributions as distinct as possible, we are primarily measuring the model's performance on compounds are at least *very infrequent* in training.

We use the MCD split generated for SCAN from Keysers et al. (2020) and the Target Maximum Compound Divergence (TMCD) splits generated for GeoQuery from Shaw et al. (2021). TMCD splits extend the MCD methodology to create compositional splits of non-synthetic datasets and define atoms and compounds based on the known syntactic structure of the output logical forms.

**Datasets and Tasks**   Table 1 summarizes the 5 compositional splits we use in this paper. We also compared with standard splits (*std*) where the training and test come from the same distribution.

Table 1: The datasets and their compositional splits used in this paper. We classify compositional splits into two categories: general CG splits where the training and test sets differ in compound distribution (*cd*) or in a range of compositional properties (*cg*), and a special type of length CG splits where the training and test sets differ in sentence length (*len*).

| Category | Dataset | Shorthand | Train | Test |
|---|---|---|---|---|
| CG General | COGS | $\text{COGS}_{cg}$ | $\text{COGS}_{\text{BASE}}$ | $\text{COGS}_{\text{CG}}$ |
| | GeoQuery | $\text{GEO}_{cd}$ | $\text{GEO}_{\text{TMCD1}}$ | $\text{GEO}_{\text{TMCD2}}$ |
| | SCAN | $\text{SCAN}_{cd}$ | $\text{SCAN}_{\text{MCD1}}$ | $\text{SCAN}_{\text{MCD2}}$ |
| CG Length | GeoQuery | $\text{GEO}_{len}$ | $\text{GEO}_{\text{SHORT}}$ | $\text{GEO}_{\text{LONG}}$ |
| | SCAN | $\text{SCAN}_{len}$ | $\text{SCAN}_{\text{SHORT}}$ | $\text{SCAN}_{\text{LONG}}$ |

As an example for our setting in §2, taking COGS as our pre-finetuning task and GeoQuery as our target task, we learn from the CG General split of COGS and transfer to the CG General split of GeoQuery. In this example, $\text{COGS}_{\text{BASE}}$ is the distribution $s$ and $\text{COGS}_{\text{CG}}$ is the distribution $\tilde{s}$. $\text{GEO}_{\text{TMCD1}}$ is our fine-tuning distribution $q$ and $\text{GEO}_{\text{TMCD2}}$ is our final test distribution $\tilde{q}$.

COGS (Kim & Linzen, 2020) is a synthetic semantic parsing dataset generated from templates. The inputs are English sentences and the outputs are corresponding logical forms inspired by $\lambda$-calculus, *e.g.*, *A dog ate the cake* $\rightarrow$ *cake$(x_4)$; dog$(x_1)$ AND eat.agent$(x_2, x_1)$ AND eat.theme$(x_2, x_4)$.

COGS has two components: a training dataset, which we refer to as $\text{COGS}_{\text{BASE}}$, and a compositional generalization (CG) dataset $\text{COGS}_{\text{CG}}$. Kim & Linzen (2020) show that the performance of neural models trained on $\text{COGS}_{\text{BASE}}$ degrades significantly when they are applied to $\text{COGS}_{\text{CG}}$. $\text{COGS}_{\text{CG}}$ includes two kinds of CG challenges: lexical and structural. In *lexical CG*, familiar words need to be interpreted in new syntactic positions; for example, in $\text{COGS}_{\text{BASE}}$ the word *hedgehog* may occur only in the subject position, in $\text{COGS}_{\text{CG}}$ it would need to be interpreted as as object. By contrast, *structural CG* involves new combinations of familiar syntactic structures. For example, in $\text{COGS}_{\text{BASE}}$, prepositional phrases only modify objects (*Noah ate the cake on the plate*), whereas in $\text{COGS}_{\text{CG}}$ they modify subjects (*The cake on the plate burned*). Likewise, in $\text{COGS}_{\text{BASE}}$ prepositional phrases can only be nested once (*Ava saw the ball in the bottle on the table*), but in $\text{COGS}_{\text{CG}}$ they are nested multiple times (*Ava saw the ball in the bottle on the table on the floor*), creating longer sentences. $\text{COGS}_{\text{BASE}}$ has 24,155 examples and $\text{COGS}_{\text{CG}}$ has 21,000 examples, of which 18,000 are instances of lexical CG and 3,000 are instances of structural CG.

The GeoQuery (Zelle, 1995; Tang & Mooney, 2001) is an annotated semantic parsing dataset contains 880 natural language questions about US geography (e.g., *What states border Texas?*). We use the same pre-processing as in Shaw et al. (2021), replacing entity mentions with placeholders in the Functional Query Language (FunQL; Kate et al. 2005) output representations. The input *what states border m0*, for example, is associated with the output `answer(intersection(state, next_to_2(m0)))`. Atoms and compounds are defined over the FunQL expressions. We adopt the length and TMCD splits from Shaw et al. (2021). For the length split $\text{GEO}_{len}$, the training set $\text{GEO}_{\text{SHORT}}$ consists of examples with shorter inputs than the test set $\text{GEO}_{\text{LONG}}$. We refer to the training and test set for the TMCD split as $\text{GEO}_{\text{TMCD1}}$ and $\text{GEO}_{\text{TMCD2}}$. Both splits divide GeoQuery equally into two sets of 440 examples.

The SCAN dataset (Lake & Baroni, 2018) consists of over 20,000 natural language navigation commands and corresponding "action sequences", *e.g. jump twice* $\rightarrow$ JUMP JUMP. SCAN is not a semantic parsing dataset but is commonly used as a diagnostic dataset for evaluating compositional generalization. We adopt the length split ($\text{SCAN}_{len}$) and MCD split ($\text{SCAN}_{cd}$) introduced by Keysers et al. (2020).

## 4 PROPOSED METHOD

**Intuition and Main Idea** As mentioned in §2, our goal is to improve compositional generalization on our target task, where the training distribution is $q$ and the evaluation distribution $\tilde{q}$. The challenge is to leverage the information in $s$ and $\tilde{s}$ during the pre-finetuning stage such that the model, when fine-tuned on $q$, generalizes better on $\tilde{q}$. This contrasts with a standard supervised learning approach,

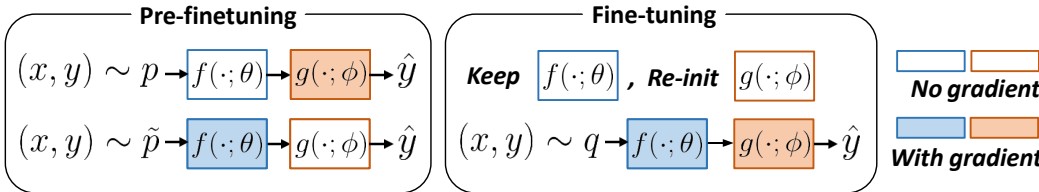

Figure 1: The DUEL updates. The parameters in the representation function $f(\cdot)$ and the task head $g(\cdot)$ are updated using different data distributions in pre-finetuning.

which would collapse together the examples from $s$ and $\tilde{s}$ during pre-finetuning, discarding the information about how examples are divided between $s$ and $\tilde{s}$—information that, we hypothesize, can be useful for encouraging compositional generalization on the target task.

We assume a neural encoder-decoder architecture (Cho et al., 2014). Our proposed method, DUEL, iteratively updates the parameters of the encoder on examples from $\tilde{s}$, while keeping the decoder fixed, and then updates the decoder parameters on examples from $s$, while keeping the encoder fixed. The encoder updates are then retained prior to fine-tuning on $q$, while the decoder updates are discarded (as the pre-finetuning and the target tasks are different). In contrast to standard fine-tuning, DUEL encourages the encoder to learn to represent the input sequences in a way that facilitates compositional generalization, which we hypothesize will transfer across domains and persist through fine-tuning.

**Details**    For encoder-decoder architectures, the model's parameters naturally partition into two sets, $\theta$ for the encoder and $\phi$ for the decoder, where $\theta$ parametrizes the representation $f(x; \theta)$ of the input $x$ and $\phi$ parametrizes the "task head" $g(\cdot)$, which uses that representation to accomplish a task such as classification or semantic parsing. Taken together, the model's output is $p(y|x; \theta, \phi) = g(f(x; \theta); \phi)$.

Given a pre-finetuning task where both $s$ and $\tilde{s}$ are given, our goal is to drive the learner to learn the optimal representation and task head parameters $(\theta^*, \phi^*)$ such that $g(f(x; \theta^*); \phi^*)$ performs the best on both distributions. To achieve this, we design an iterative dueling game that the learner plays with the two distributions.

Suppose the learner's parameters are currently $(\theta, \phi)$. To perform optimally on $\tilde{s}$, we would like to use the task head $g(\cdot; \phi)$ without further training it, *i.e.*, perform well in an "zero-shot" setting; this setting directly anticipates how learning would proceed for the target task, where we will be training only on $q$, without exposure to $\tilde{q}$. Since we are holding the task head $g(\cdot; \phi)$ fixed, we need to optimize the representation $f(\cdot; \theta)$ such that it models $\tilde{s}$ appropriately. To achieve this desiderata, we update the representation parameters iteratively for at most $T_{\text{inner}}$ steps, according to

$$\theta \leftarrow \theta + \alpha \frac{1}{N} \nabla_\theta \sum_{(x,y) \in B_{\tilde{s}}} \log p(y|x; \theta, \phi) \qquad (1)$$

where $B_{\tilde{s}}$ is a batch of $N$ samples from $\tilde{s}$ and $\alpha$ is the step size.[2]

Updates to $\phi$ follow the reverse logic. The representation function that results from updating $\theta$ defines a prior on how $s$ should be represented. When updating the task head parameters $\phi$, our goal is to have the task head perform optimally conditioned on this prior. Concretely, we hold $\theta$ fixed and apply iteratively at most $T_{\text{inner}}$ steps to $\phi$:

$$\phi \leftarrow \phi + \alpha \frac{1}{N} \nabla_\phi \sum_{(x,y) \in B_s} \log p(y|x; \theta, \phi) \qquad (2)$$

where $B_s$ is a batch of samples from $s$. Crucially, we alternate between these two types of updates, such that $\tilde{s}$ informs the learner about the compositional inductive bias that should be incorporated into the representation $f$, and $s$ teaches the learner how to use that representation to perform the task (here semantic parsing).

---

[2]Eqns. (1-3) only aim at illustrating the computation of gradient. We use the Adam optimizer in the implementation.

---

**Algorithm 1:** DUEL

---

**Require:** Data sets $s$, $\tilde{s}$; Learning rate $\alpha$; Batch size $N$; Outer loop iterations $T_{\text{outer}}$; Inner loop
iterations $T_{\text{inner}}$; Inner loop early stopping criteria $T_{\text{patience}}$; Outer loop early stopping criteria
$T_{\min}$;
Initialize model parameters $\theta, \phi$
$i \leftarrow 0$
**while** $i < T_{\text{outer}}$ **do**
    $j \leftarrow 0$
    **while** $j < T_{\text{inner}}$ **do**
        Sample a batch $B_s$ from $s$
        Compute loss: $\mathcal{L}(\theta, \phi, B_s) = \frac{1}{N} \sum_{(x,y) \in B_s} - \log p(y \mid x; \theta, \phi)$
        Update parameters: $\phi \leftarrow \phi - \alpha \cdot \nabla_\phi \mathcal{L}(\theta, \phi, B_s)$
        $j \leftarrow j + 1$
        **if** AccuracyDecreases*($\theta$, $\phi$, $\tilde{s}$, $T_{\text{patience}}$)* **then**
            | Early stop: **break**
        **end**
    **end**
    $k \leftarrow 0$
    **while** $k < T_{\text{inner}}$ **do**
        Sample a batch $B_{\tilde{s}}$ from $\tilde{s}$
        Compute loss: $\mathcal{L}(\theta, \phi, B_{\tilde{s}}) = \frac{1}{N} \sum_{(x,y) \in B_{\tilde{s}}} - \log p(y \mid x; \theta, \phi)$
        Update parameters: $\theta \leftarrow \theta - \alpha \cdot \nabla_\theta \mathcal{L}(\theta, \phi, B_{\tilde{s}})$
        $k \leftarrow k + 1$
        **if** AccuracyDecreases*($\theta$, $\phi$, $s$, $T_{\text{patience}}$)* **then**
            | Early stop: **break**
        **end**
    **end**
    **if** $k < T_{\min}$ **then**
        | Early stop outer loop: **break**
    **end**
    $i \leftarrow i + 1$
**end**
**Return:** Model parameters $\theta, \phi$

---

We contrast our approach with the standard supervised learning approach, where the two distributions
are **merged** together as $s \cup \tilde{s}$, ignoring the compositional split:

$$(\theta, \phi) \leftarrow (\theta, \phi) + \eta \frac{1}{N} \nabla_{(\theta, \phi)} \sum_{(x,y) \sim B_{s \cup \tilde{s}}} \log p(y|x; \theta, \phi) \qquad (3)$$

DUEL is used for pre-finetuning. When fine-tuning the model on the target task $q$, we retain the
representation component $f(\cdot; \theta)$ and re-initialize the task head $g(\cdot; \phi)$. Both $\theta$ and $\phi$ are then updated,
as is standard, to optimize the loss function on $q$.

DUEL is illustrated schematically in the pre-finetuning panel of Fig. 1. The pseudocode is listed in
Algorithm 1. The algorithm contains one outer loop for $T_{\text{outer}}$ rounds (with possible early stopping),
which alternates between inner loops updating $\theta$ and $\phi$. The early-stopping monitoring function
AccuracyDecreases takes as arguments the current model parameters $\theta$ and $\phi$, a data distribution
to use for evaluation ($s$ or $\tilde{s}$), and the maximum patience $T_{\text{patience}}$ for early-stopping. This function
returns `true` if accuracy on the dataset in question has not improved for $T_{\text{patience}}$ consecutive steps.
To terminate the outer loop (thus, the algorithm), we keep track how many steps the inner update for
$\theta$ take. If the number steps is smaller than a preset threshold $T_{\min}$, we conclude that the algorithm has
converged to the desired representation since the difference in representing $s$ and $\tilde{s}$ is small, requiring
little adaptation. We do not use the same logic to limit how long it takes to optimize the task head
parameters $\phi$, conditioned on the representation, as the goal is to use the derived representation to
arrive at a strong performance.

# 5 EXPERIMENTS

## 5.1 SETTINGS

**Datasets and Tasks** We use the datasets described in Table 1. The target tasks always come from one of the compositional splits: $COGS_{cg}$, $GEO_{cd}$, $SCAN_{cd}$, or $GEO_{len}$. For the pre-finetuning tasks, we consider two configurations: (1) standard splits for COGS, GeoQuery and SCAN; (2) compositional splits: $COGS_{cg}$, $GEO_{cd}$, $SCAN_{cd}$, $SCAN_{len}$.

**Models** We use two neural architectures, both encoder-decoder models based on the Transformer architecture (Vaswani et al., 2017). The first one, which we refer to as BERT2SEQ, uses the BERT encoder $BERT_{SMALL}$ (Devlin et al., 2019), with 4 Transformer layers, to learn the representation $f(x;\theta)$, followed by a Transformer-based sequence-to-sequence model with 2 encoder layers and 2 decoder layers, as the task head $g(\cdot;\phi)$.[3] We initialize $f(\cdot)$ with the pre-trained $BERT_{SMALL}$ (Turc et al., 2019).

We also experimented with the much larger T5-Base model (Raffel et al., 2020). We take the T5 encoder as $f(x;\theta)$ and the T5 decoder as the task head $g(\cdot;\phi)$; the encoder and encoder have 12 Transformer layers each. We initialize them with the pre-trained T5 model. In both pre-finetuning and fine-tuning, we generate task prompts by prepending a task tag to each input sentence (following Raffel et al. 2020). We provide detailed examples of the task prompts for each task in Appendix B.

When using BERT2SEQ, our main results are obtained by initializing the task heads randomly. For T5, we initialize the task heads with the pre-trained T5 decoder, for both pre-finetuning and fine-tuning.

**Baselines** We compare DUEL to two baselines: (1) NONE, where we fine-tune on the target task (*i.e.*, $q$) directly, without pre-finetuning; (2) MERGED, pre-finetuning on $s \cup \tilde{s}$ using eq. (3) and then fine-tuning on the target task. As mentioned above, MERGED ignores the compositional splits and conducts pre-finetuning using the standard splits (cf. Table 1).

## 5.2 DUEL LEARNS TRANSFERABLE COMPOSITIONAL GENERALIZATION INDUCTIVE BIAS

**Main Results** Table 2 shows the results of our experiments taking the CG General splits as the target splits. The metric we report is exact match accuracy over output sequences. For COGS, we report accuracy separately for the lexical and structural generalization cases, as our models' performance differed dramatically between those two types of cases. On SCAN, the NONE baseline (no pre-finetuning) matches the average accuracy 15.4% reported by Conklin et al. (2021).

Pre-finetuning substantially improves over the baselines, across the board. In particular, pre-finetuning on $SCAN_{cd}$ leads to an accuracy of 57.8% on $GEO_{TMCD2}$, surpassing — with a neural-only approach — the previous state-of-the-art of 56.6%, which was established by NQG-T5, a hybrid model that combines a grammar-based approach with T5 (Shaw et al., 2021). The detailed comparison to other existing approaches is provided in Appendix D.1

**Pre-finetuning on a compositional split is essential** Do the improvements we observe in Table 2 specifically reflect the contribution of pre-finetuning on compositional splits, or can they be attributed to pre-finetuning more generally? Table 2 reports the performance of the MERGED baseline, *i.e.*, pre-finetuning on the merged dataset $s \cup \tilde{s}$, without a compositional split. Comparing to the NONE (no pre-finetuning) rows in Table 2, we observe that pre-finetuning improves performance on the target tasks even without using DUEL or the compositional split. However, we see bigger gains from using DUEL on compositional splits (DUEL rows with CG General category in Table 2), especially for the target tasks $GEO_{cd}$ and $SCAN_{cd}$. On $COGS_{cg}$, the gain of using DUEL on compositional splits is less pronounced. This reflects two facts: first, that lexical generalization is not very challenging when the encoder uses a pre-trained model with strong lexical representations, such that performance is close to being saturate even without pre-finetuning; and second, that the structural generalization cases in COGS remain too difficult for purely neural models, even with DUEL.

---

[3] The model is larger than the baseline Transformer model from Kim & Linzen (2020), which has the same architecture as $g(\cdot)$ alone. The model trained by Kim & Linzen (2020) does not use pre-finetuning and is outperformed by our model in the same setup from a pre-trained $BERT_{SMALL}$.

Table 2: Accuracy on the target tasks after pre-finetuning on other tasks. Note that the MERGED baseline is the same for the Standard and CG General splits as it uses the union of two distributions. For COGS, we report lexical and structural CG accuracy separately (lexical/structural).

| Model | Pre-fine tuning | Category | GEO →COGS$_{cg}$ | SCAN | COGS →GEO$_{cd}$ | SCAN | COGS →SCAN$_{cd}$ | GEO |
|---|---|---|---|---|---|---|---|---|
| BERT2SEQ | NONE | - | 50.8/0.2 | 50.8/0.2 | 34.3 | 34.3 | 1.3 | 1.3 |
| | MERGED | Standard | 71.2/0.7 | 66.9/0.8 | 35.9 | 36.1 | 3.2 | 3.0 |
| | DUEL | Standard | 63.4/1.0 | 61.5/0.3 | 35.2 | 35.7 | 2.9 | 3.1 |
| | DUEL | CG General | 75.5/1.1 | 73.2/1.1 | 36.9 | 37.7 | 4.8 | 5.5 |
| T5-Base | NONE | - | 93.7/2.5 | 93.7/2.5 | 52.5 | 52.5 | 15.4 | 15.4 |
| | MERGED | Standard | 95.2/4.2 | 94.3/3.4 | 52.5 | 54.3 | 18.7 | 16.9 |
| | DUEL | Standard | 94.9/3.1 | 94.1/3.2 | 51.1 | 51.6 | 19.2 | 15.7 |
| | DUEL | CG General | **95.4/4.5** | **94.9/3.9** | **53.8** | **57.8** | **20.2** | **21.3** |

Table 3: Results on length generalization (GEO$_{SHORT}$ →GEO$_{LONG}$) from different pre-finetuning tasks. Numbers in parentheses show the improvement of DUEL from MERGED. COGS ∪ SCAN denotes pre-finetuning with COGS$_{cg}$ and SCAN$_{len}$ together with two separate decoders.

| Model | Pre-finetuning | SCAN$_{std}$ | SCAN$_{cd}$ | COGS$_{cg}$ | SCAN$_{len}$ | COGS ∪ SCAN |
|---|---|---|---|---|---|---|
| BERT2SEQ | NONE | | | 16.1 | | |
| | MERGED | **16.1** | 16.1 | 16.8 | 16.1 | 17.0 |
| | DUEL | 15.9 (-0.2) | **16.8** (+0.7) | **17.7** (+0.9) | **18.6** (+2.5) | **18.9** (+1.9) |
| T5-Base | NONE | | | 38.6 | | |
| | MERGED | 39.1 | 39.1 | 40.9 | 39.1 | 41.4 |
| | DUEL | **39.3** (+0.2) | **40.0** (+0.9) | **43.0** (+2.1) | **45.0** (+5.9) | **46.4** (+5.0) |

Within Table 2, it is clear that DUEL does not make a major difference from MERGED when restricted to standard splits . This is reasonably expected as the DUEL procedure reduces to the MERGED when the data distribution $s$ and $\tilde{s}$ are exactly the same for the pre-finetuning task (cf. eq.(3) to eq. (1,2)).

In summary, the results so far suggest that DUEL is most effective when used in conjunction with pre-finetuning tasks with compositional splits.

## 5.3 WHEN WILL DUEL WORK BEST?

We hypothesize DUEL works best when the pre-finetuning tasks and the target tasks share strong similarity in their compositional splits. We gain further insights about how effective DUEL is by restricting our study to a special type of compositional generalization - the CG Length generalization. Table 3 reports the results when the compsitional split in the target task GEO$_{len}$ is to generalize from shorter texts to longer ones. Without pre-finetuning, the accuracy on GEO$_{len}$ is 38.6% with T5-Base, and 16.1% with BERT2SEQ. The state-of-the-art results on this task is 52.2% from the NQG-T5 model (Shaw et al., 2021).

As before, pre-finetuning improves from that baseline. SCAN$_{len}$ with DUEL improves the most for both neural models. Intuitively, this is sensible as SCAN$_{SHORT}$ also consists of examples with shorter texts than those in the test set SCAN$_{LONG}$: on average, the input lengths are 7.0 versus 8.2 on SCAN$_{SHORT}$ and SCAN$_{LONG}$ respectively, while the output lengths significantly vary from 10.8 to 30.0. In this case, DUEL helps extract the inductive bias for generalizing on length difference.

Our experiments also show that COGS$_{cg}$ works better than SCAN$_{cd}$. The compositional splits in COGS$_{cg}$ are multiple typed and not all about length difference. Nonetheless, in the structural challenge portion of the COGS$_{cg}$, the average text length is 61 in COGS$_{CG}$, compared to 22 in COGS$_{BASE}$. On the other hand, SCAN$_{MCD1}$ and SCAN$_{MCD2}$ have average text lengths of 7.0 and 7.7, respectively. The SCAN$_{std}$, which is the standard split, does not improve as it is not a compositional split.

Table 4: Accuracy of T5-Base on COGS's structural challenge, with different pre-finetuned tasks

| Pre-finetuning | SCAN$_{cd}$ | GEO$_{cd}$ | 1×COGS_VAR$_{cg}$ | 5×COGS_VAR$_{cg}$ | 10× COGS_VAR$_{cg}$ |
|---|---|---|---|---|---|
| MERGED | 3.4 | 4.2 | 4.9 | 5.1 | 4.8 |
| DUEL | 3.9 | 4.5 | 4.9 | 5.2 | 5.2 |

Also, pre-finetuning on two tasks shows the best performance, which implies the potential of our method to learn a general compositional generalization strategy with multiple pre-finetuning tasks.

### 5.4 HOW MUCH CAN WE IMPROVE PERFORMANCE OF COGS BY GENERATING VERY SIMILAR PRE-FINETUNING TASKS?

While DUEL performed well on the SCAN and GeoQuery target tasks, performance on COGS was less impressive, especially on the structural generalization cases (Table 2). It has recently been shown that the challenges presented by COGS can be addressed using symbolic methods, albeit ones that are informed by detailed knowledge of the generative process that was used to create the dataset (Liu et al., 2021). Can we make additional progress on this dataset using a generic sequence-to-sequence neural model?

We hypothesized that the pre-finetuning splits we used before fine-tuning on COGS were insufficiently similar to the split required for COGS (unlike, for example, the SCAN length split that correspond to the generalization behavior required for GEO$_{len}$). To test this hypothesis, as a lower bound on DUEL's ability to extract inductive bias for compositional generalization, we designed a new experiment such that the pre-finetuning tasks will have the **same** compositional splits as the COGS.

Concretely, we created ten COGS variants with the same type of splits as COGS$_{cg}$; the only difference between the variants was that each of them used a different lexicon. To create these variants, we first used the SpaCy Python library to identify 3 part-of-speech classes: PROPN, NOUN and VERB. For each proper noun, we selected 5 different alternatives. For each noun and verb, we selected all the synonyms and antonyms from WordNet that had the same part-of-speech. To create a single COGS variant, for each word in the COGS vocabulary, we built a 1-to-1 replacement mapping by randomly selecting one alternative, and generated new sentences by the mapping. The following pair illustrates the mapping with an example (and more in Appendix C).:

> **COGS:** `Emma ate the ring beside a bed.` ⟶
> **COGS_VAR:** `Trudy consumed the hoop beside a layer.`

The lexicons of the variants and the original COGS were completely disjoint in those part-of-speech classes; other words (in particular function words were the same across all variants. We use COGS_VAR$_{cg}$ to denote the CG split of a single COGS variant. For pre-finetuning, we used 1, 5 or 10 variants.

The results reported in Table 4 suggest that pre-finetuning with the same type of compositional generalization split only leads to a very minor improvement in accuracy, from 4.5% to 5.2%. Note that the pre-finetuning tasks have an average accuracy of 79.3% on the structural generalization portions of those tasks. Our method is thus surprisingly ineffective in this case, even when the pre-finetuning task is very similar in structure to the target task. For future research, we plan to investigate the reasons of the limited improvement and approaches for extracting stronger inductive bias.

## 6 RELATED WORK

**Transfer Learning** A large body of work has attemped to leveraged multi-task learning to endow a model with an inductive bias that improves generalization on a main task of interest (Caruana, 1998; Bakker & Heskes, 2003; Raffel et al., 2020), with recent work in NLP sharing our focus on neural networks (Sogaard & Goldberg 2016; Hashimoto et al. 2016; Swayamdipta et al. 2018; for a review, see Ruder 2017). Intermediate training of pre-trained sentence encoders on a task or a set of tasks that are related to the task of interest has been advocated, among others, by Phang et al. (2018)

and Aghajanyan et al. (2021). Gururangan et al. (2020) craft a training pipeline where a pre-trained language model is adapted to domain-specific and task-specific ones. There are also empirical studies of transferrability among tasks (Vu et al., 2020). Our work differs from most of the work in this area in that our goal is to learn to generalize compositionally from a training distribution to a test one; as such we provide the model with a strategic training and **test set** split drawn from a pre-finetuning task, which together illustrate the desired generalization behavior that we intend for the model to transfer to a target task. This approach bears some resemblance to work on learning-to-learn or meta-learning (Thrun & Pratt, 1998; Gu et al., 2018), and a recent study on learning low-resource tasks (Chen et al., 2020a).

**Compositional Generalization** A number of approaches have been explored to achieve compositional generalization in neural models for NLP. Specifically, recent work has proposed new or modified model architectures (Li et al., 2019; Russin et al., 2019; Gordon et al., 2020; Liu et al., 2020; Nye et al., 2020; Chen et al., 2020b; Zheng & Lapata, 2020; Oren et al., 2020; Herzig & Berant, 2020; Shaw et al., 2021; Yin et al., 2021). Furrer et al. (2020) compared several of these architectures with general-purpose pre-trained models such as T5. Other work has explored intermediate representations (Herzig et al., 2021) and compositional data augmentation procedures (Andreas, 2020; Akyürek et al., 2020).

Our approach is most closely related to work that applies meta-learning to compositional generalization (Conklin et al., 2021; Oren et al., 2021; Lake, 2019). Conklin et al. (2021) focus on using tasks that are similar to the target task to regularize the learning of neural models. By systematically generating compositional splits on the synthetic dataset SCAN, Oren et al. shows that the success of meta-learning depends on the specific splits that are presented to the learner (Oren et al., 2021). Our work complements these studies by showing that it is possible to transfer an inductive bias learned on one task to a substantially different task.

## 7 CONCLUSION

In this paper, we studied acquiring compositional generalization skills through a pre-finetuning task, where a compositional split is available, and transferring those skills to a target task where such a compositional split does not exist. We proposed DUEL, a simple method for this purpose. The underlying intuition of this method is that we would like to learn an input representation that incorporates a compositional inductive bias, and train it separately from a task head that performs a specific task. We demonstrated the effectiveness of this method on 3 semantic parsing tasks. In particular, using the COGS and SCAN synthetic tasks as pre-finetuning tasks, we were able to achieve high compositional generalization accuracy on two splits of the non-synthetic GeoQuery dataset: the Target Maximum Compound Divergence split $GEO_{cd}$ and the length split $GEO_{len}$. For $GEO_{cd}$, our approach beats the state-of-the-art, established by a hybrid neurosymbolic model (Shaw et al., 2021); for $GEO_{len}$, we substantially improved over the baselines.

**Reproducibility Statement** To ensure reproducibility, we describe the datasets and their train/test splits for our experiments in Section 3 and Section 5.1. We explain our algorithm with pseudocode in Section 4 Algorithm 1. We also describe the model architecture and baselines in Section 5.1. Further, we provide detailed hyper-parameters and hyper-parameter tuning methods in Appendix A. We give examples of preprocessed inputs to T5 for each dataset in Appendix B. We will release the code after the anonymity period at `https://www.anonymous.com`.

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

# Appendix

In this supplementary material, we provide details omitted in the main text. The content is organized as what follows:

## A    IMPLEMENTATION DETAILS

### A.1    HYPER-PARAMETERS IN FINE-TUNING

First, we list the hyper-parameters in fine-tuning for each model and task.

For COGS, we fine-tune for a maximum of 30k steps with a learning rate of $1e^{-4}$. Following Kim & Linzen (2020), we evaluate every 500 iterations on the dev set of COGS and early-stop if the accuracy on the dev set has not improved for 5 consecutive step. The batch size is 128 on T5-base and 512 on BERT2SEQ.

For GeoQuery, it has no dev set. Therefore, we tune hyper-parameters on the standard split of the GeoQuery dataset. We then use the same hyper-parameters for the $GEO_{cd}$ and $GEO_{len}$ splits. On T5-base, we fine-tune for 5k steps with a batch size of 256 and a learning rate of $1e^{-4}$. On BERT2SEQ, we fine-tune for 3k steps with a batch size of 512 and a learning rate of $2e^{-5}$.

For SCAN, likewise, we tune hyper-parameters on the standard split of it. We then use the same hyper-parameters for the $SCAN_{cd}$ and $SCAN_{len}$ splits. On T5-base, we fine-tune on for 3.5k steps with a batch size of 128 and a learning rate of $1e^{-4}$. On BERT2SEQ, we fine-tune for 2.5k steps with a batch size of 512 and a learning rate of $1e^{-4}$.

We use the Adam optimizer with weight decay. We take the label smoothing (Müller et al., 2019) with weight factor 0.1 on BERT2SEQ but not T5-base.

Next, we provide the computation time for fine-tuning on 1 TITAN Xp GPU. For COGS, the average fine-tuning time is about 24 hours on T5-base and 4 hours on BERT2SEQ. For GeoQuery, about 2 hours on T5-base and 0.4 hour on BERT2SEQ. For SCAN, about 3 hours on T5-base and 0.5 hours on BERT2SEQ.

### A.2    VALIDATION DATASET FOR HYPER-PARAMETER TUNING

As we train on the whole dataset with DUEL during pre-finetuning, we are not able to directly tune hyper-parameters for DUEL. In order to validate the learned representation is good at OOD generalization from $s$ to $\tilde{s}$ without accessing the test data in $\tilde{s}$, we hold out a validation dataset for each task from $\tilde{s}$. We then take the pre-finetuned model, re-initialize the task-head, fine-tune it on $s$ and test on the hold-out set of $\tilde{s}$. We select the hyper-parameters of DUEL based on the test accuracy on the hold-out set.

For COGS, we create a validation dataset by holding a subset of 3000 examples from $COGS_{CG}$. Likewise, the validation datasets of GeoQuery and SCAN hold out 20% of the data from $GEO_{TMCD2}$ and $SCAN_{MCD2}$, respectively.

### A.3    HYPER-PARAMETERS IN PRE-FINETUNING

In pre-finetuning, we choose the maximum outer loop iterations $T_{outer}$ as 10, the maximum inner loop iterations $T_{inner}$ as 30k. For efficiency, we evaluate every 500 iterations and set the early-stopping

patience for the inner loop as 1. We select a learning rate $\alpha$ from $[1e^{-5}, 2e^{-5}, 1e^{-4}]$ for all tasks and all models.

The batch size is different from each model. For T5-base, we select a batch size of 32 from [8, 16, 32] for all tasks. For BERT2SEQ, we select a batch size of 512 from [256, 512, 1024].

The early-stopping minimum iterations $T_{\min}$ for the outer loop is different for each model and task. For COGS, $T_{\min} = 3000$ on T5-base and $T_{\min} = 2000$ on BERT2SEQ. For GeoQuery, $T_{\min} = 1000$ on both T5-base and BERT2SEQ. For SCAN, $T_{\min} = 2000$ on T5-base and $T_{\min} = 1000$ on BERT2SEQ. We use the same optimizer and label smoothing method in fine-tuning as in pre-finetuning.

Next, we provide the computation time for DUEL on 1 TITAN Xp GPU. When pre-finetuning on COGS, the average running time is about 60 hours on T5-base and 24 hours on BERT2SEQ. For GeoQuery, about 24 hours on T5-base and 2 hours on BERT2SEQ. And for SCAN, about 30 hours on T5-base and 3 hours on BERT2SEQ.

## B  T5 PROMPT DETAILS

In this section, we provide examples of T5 prompt for each task in our experiment. Our design of task prompt follows the standard way from Raffel et al. (2020)

### B.1  COGS

**Original input:**  `Emma ate the ring beside a bed .`

**Prompt input:**  `cogs:  Emma ate the ring beside a bed .`

### B.2  COGS_VAR

**Original input:**  `The tiddler appreciated that Kim measure that a cooky was slid .`

**Prompt input:**  `cogs:  The tiddler appreciated that Kim measure that a cooky was slid .`

### B.3  GEOQUERY

**Original input:**  `what is the largest state capital in population`

**Prompt input:**  `geo:  what is the largest state capital in population`

### B.4  SCAN

**Original input:**  `run around right after look opposite left`

**Prompt input:**  `scan:  run around right after look opposite left`

## C  MORE EXAMPLES ON COGS_VAR VS COGS

For each COGS example listed following, we give three examples from the corresponding COGS_VAR.

**COGS input sentence 1:**  `The sailor dusted a boy .`

**COGS output program 1:** `*sailor(x_1); dust.agent(x_2, x_1) AND dust.theme(x_2, x_4) AND boy(x_4)`

**COGS_VAR input sentences 1:**

The crewman scattered a female_child .

The boater dotted a male_child .

The leghorn dotted a son .

**COGS_VAR output programs 1:**

*crewman(x_1); scatter.agent(x_2, x_1) AND scatter.theme(x_2, x_4) AND female_child(x_4)

*boater(x_1); dot.agent(x_2, x_1) AND dot.theme(x_2, x_4) AND male_child(x_4)

*leghorn(x_1); dot.agent(x_2, x_1) AND dot.theme(x_2, x_4) AND son(x_4)

**COGS input sentence 2:** The cookie was passed to Emma .

**COGS output program 1:** *cookie(x_1); pass.theme(x_3, x_1) AND pass.recipient(x_3, Emma)

**COGS_VAR input sentences 2:**

The cooky was faded to Joan .

The cooky was overstepped to Trudy .

The cooky was eliminated to Kim .

**COGS output programs 2:**

*cooky(x_1); fade.theme(x_3, x_1) AND fade.recipient(x_3, Joan)

*cooky(x_1); overstep.theme(x_3, x_1) AND overstep.recipient(x_3, Trudy)

*cooky(x_1); eliminate.theme(x_3, x_1) AND eliminate.recipient(x_3, Kim)

**COGS input sentence 3:** Zoe thought that a hippo cleaned .

**COGS output program 3:** think.agent(x_1, Zoe) AND think.ccomp(x_1, x_5) AND hippo(x_4) AND clean.agent(x_5, x_4)

**COGS_VAR input sentences 3:**

Larry cerebrated that a Hippo housecleaned .

Dana guessed that a hippopotamus picked.

Ronnie retrieved that a hippopotamus make_cleaned .

**COGS output programs 3:**

cerebrate.agent(x_1, Larry) AND cerebrate.ccomp(x_1, x_5) AND Hippo(x_4) AND houseclean.agent(x_5, x_4)

guess.agent(x_1, Dana) AND guess.ccomp(x_1, x_5) AND hippopotamus(x_4) AND pick.agent(x_5, x_4)

retrieve.agent(x_1, Ronnie) AND retrieve.ccomp(x_1, x_5) AND hippopotamus(x_4) AND make_clean.agent(x_5, x_4)

# D    ADDITIONAL EXPERIMENTAL RESULTS

In this section, we provide the experimental results as well as the standard deviation of DUEL omitted in the main text.

Table 5: Best accuracy on the target tasks $COGS_{cg}$, $GEO_{cd}$ and $SCAN_{cd}$

| Method | $COGS_{cg}$ | $GEO_{cd}$ | $SCAN_{cd}$ |
|---|---|---|---|
| SBSP | - | 49.2 | **100** |
| NQG-T5 | - | 56.6 | **100** |
| MAML | 92.7/3.1 | - | 15.9 |
| DUEL | **95.4/4.5** | **57.8** | 21.3 |

## D.1    COMPARISON WITH THE EXISTING WORKS

Several existing works leverage the specific designs of the datasets to solve SCAN (Andreas, 2020; Li et al., 2019; Russin et al., 2019), most of the improvements do not transfer to non-synthetic tasks. We compare to two methods that solve SCAN but can also transfer to non-synthetic tasks, SpanBasedSP (Herzig & Berant, 2020) and NQG-T5 (Shaw et al., 2021). Also, we compare to a recently proposed model-agnostic meta-learning approach (Conklin et al., 2021).

We use the original released code to run the experiments for SBSP (SpanBasedSP) and MAML. For SBSP, we use the BERT-BASE model as in their work; For MAML, we replace their 2-layer Transformer encoder and 2-layer Transformer decoder with the larger model T5-Base Encoder-Decoder for a fair comparison.

DUEL outperforms MAML on $SCAN_{cd}$ and $COGS_{cg}$, and outperforms SpanBasedSP substantially on $GEO_{cd}$. As expected, it falls behind SpanBasedSP and NQG-T5 on $SCAN_{cd}$.

Table 6: Average input length of correctly predicted examples on length generalization ($GEO_{SHORT}$ →$GEO_{LONG}$) from different pre-finetuning tasks. Number of correct examples show in parentheses.

| Model | Pre-finetuning | $SCAN_{std}$ | $SCAN_{cd}$ | $COGS_{cg}$ | $SCAN_{len}$ |
|---|---|---|---|---|---|
| | NONE | | 7.633 (71) | | |
| BERT2SEQ | MERGED | 7.634 (71) | 7.634 (71) | 7.642 (74) | 7.634 (71) |
| | DUEL | 7.620 (70) | 7.651 (74) | 7.657 (78) | 7.693 (81) |
| | NONE | | 8.876 (170) | | |
| T5-Base | MERGED | 8.872 (172) | 8.872 (172) | 8.899 (180) | 8.872 (172) |
| | DUEL | 8.865 (173) | 8.960 (176) | 8.968 (189) | 8.985 (198) |

## D.2    WHAT TRANSFERS BETWEEN DATASETS?

The ability to transfer across datasets is related to what exactly the model is learning by DUEL during the pre-finetuning stage. Our hypothesis is that DUEL might restrict the encoder to learn to map structures compositionally, bits and pieces, instead of overfitting on arbitrary complex "structural noises". In the light of this hypothesis, the DUEL pre-finetuning regularizes the encoder to represent structural information in a more decomposable way, instead of "contextually interdependent". Suppose the input sentence is "jump right after walk right". Ideally, we would want the encoder to pick up the cue that "after" signifies to represent "jump right" and "walk right" separately. However, for an end-to-end model, the representation could be biased to couple these two shorter phrases together to map to the desired output sequence.

We have not come up with a definitive way to test the theory, instead, we added some indirect evidences to support the theory. We compare the input length of correctly predicted examples in $GEO_{len}$ between training with DUEL and MERGED in Table 6. The results indicate that the improvement mostly comes from the longer inputs, which means the encoder is being regularized on the abstract notion of learning better from shorter structures and supports our hypothesis.

Table 7: Accuracy on the target tasks after pre-finetuning on other tasks with splits in the CG General category. For COGS, we report lexical and structural CG accuracy separately (lexical/structural). All the results are averaged over 3 runs.

| Model | Pre-finetuning | GEO $\rightarrow$COGS$_{cg}$ | SCAN | COGS $\rightarrow$GEO$_{cd}$ | SCAN | COGS $\rightarrow$SCAN$_{cd}$ | GEO |
|---|---|---|---|---|---|---|---|
| BERT2SEQ | DUEL(F) | 61.4/1.0 | 58.6/1.1 | 35.8 | 36.1 | 4.4 | 4.8 |
| | DUEL(S) | 72.0/0.3 | 72.3/1.3 | 36.3 | 37.5 | 5.1 | 2.0 |
| | DUEL | 75.5/1.1 | 73.2/1.1 | 36.9 | 37.7 | 4.8 | 5.5 |
| T5-Base | DUEL(F) | 90.7/2.5 | 89.6/2.5 | 52.5 | 55.7 | 16.0 | 18.9 |
| | DUEL(S) | 94.8/2.3 | 95.1/3.6 | 53.4 | 57.5 | 20.1 | 12.7 |
| | DUEL | **95.4/4.5** | **94.9/3.9** | **53.8** | **57.8** | **20.2** | **21.3** |

## D.3 ABLATIONS ON THE DESIGN OF DUEL

We evaluate on two ablations of DUEL: (1) DUEL(F), where we keep the encoder fixed in fine-tuning; (2) DUEL(S), where we switch the train and the test component in pre-finetuning. Results are listed in Table 7.

DUEL(F) shows a large performance drop comparing with DUEL, which implies it is beneficial to finetune the encoder together with the decoder.

DUEL(S) shows minor degradation to DUEL when pre-finetuning on SCAN, likely due to its symmetric MCD split. Similarly, the COGS lexical generalization set has the same average input length as the train set and contains all the vocabs. Thus, training set is largely symmetric with its lexical generalization set, which leads to minor change in DUEL(S). However, DUEL(S) has a large gap with DUEL when pre-finetuning on GeoQuery. As the TMCD split only ensures all atoms appear in the train component GEO$_{TMCD1}$, we find it has about 30% of examples containing words that do not appear in the test component GEO$_{TMCD2}$. Thus, using GEO$_{TMCD2}$ to train the decoder and GEO$_{TMCD1}$ to train the encoder forces the encoder to model those "new" atoms. This likely causes the encoder not learning enough inductive bias, thus incurring a larger drop in performance.

## D.4 THE STANDARD DEVIATION OF DUEL

In Tables 2 and 3 we report the mean of 3 runs for accuracy of the target tasks when pre-finetuning with DUEL. The standard deviations for these runs are reported in Table 8 and 9, respectively.

Table 8: The standard deviations of the DUEL results in the CG General category in Table 2.

| Model | Pre-finetuning | GEO $\rightarrow$COGS$_{cg}$ | SCAN | COGS $\rightarrow$GEO$_{cd}$ | SCAN | COGS $\rightarrow$SCAN$_{cd}$ | GEO |
|---|---|---|---|---|---|---|---|
| BERT2SEQ | DUEL | 4.0/0.4 | 3.6/0.3 | 0.5 | 1.0 | 1.3 | 1.8 |
| T5-Base | DUEL | 1.4/0.7 | 1.5/0.2 | 0.1 | 1.4 | 0.4 | 2.8 |

Table 9: The standard deviations of the DUEL results in the CG Length category in Table 3

| Model | Pre-finetuning | SCAN$_{std}$ | SCAN$_{cd}$ | COGS$_{cg}$ | SCAN$_{len}$ | COGS $\cup$ SCAN |
|---|---|---|---|---|---|---|
| BERT2SEQ | DUEL | 1.0 | 0.5 | 0.4 | 0.5 | 0.2 |
| T5-Base | DUEL | 0.8 | 0.1 | 0.5 | 0.9 | 1.1 |

