# OpenReview forum: "Learning to Generalize Compositionally by Transferring Across Semantic Parsing Tasks"
_ICLR.cc/2022/Conference — ICLR 2022 Submitted_

### Official Review · Reviewer_K5oD · 2021-11-02

**Correctness:** 4
**Technical Novelty And Significance:** 2
**Empirical Novelty And Significance:** 3
**Recommendation:** 6
**Confidence:** 4

**Main Review:**

*Strengths*

S1) I appreciated the paper's use of a non-synthetic dataset (GeoQuery), as I feel that this is an underexplored area of work on compositional generalization which will be useful to explore how and when nets fail to generalize on real data, and how to fix them.

S2) The proposed approach seems simple and easy to implement.

S3) The experiments were overall thorough (at least in the scope of semantic parsing compositional generalization), evaluating on three different datasets and two different models.

S4) I found the demonstration of the benefits of pre-finetuning interesting and convincing.

S5) The paper was extremely clearly written, in particular the description of the method.

*Weaknesses*

W1) I wasn't totally convinced that the method works well on strong tests of compositional generalization.
- The GEO_cd and SCAN_cd splits, although they follow past work, are defined using a compound divergence method that, as the paper points out, does not ensure that compounds are completely absent (but only infrequent) in the training set.
- The COGS_cg lexical challenge seems to mostly be obviated by pre-trained representations.
- While I did find the length generalization results to be a convincing improvement and a more reasonable test of structural generalization, no method really seems to help much on the harder structural generalization test of COGS_cg, and (concerningly) even the proposed method makes no improvement in what seems to be the most a priori favorable experiment design for it, described in 5.4 (although I did appreciate including this negative result!).

W2) It's not totally clear to me why the method should enable compositional generalization in general, and I feel like it would help to strengthen the motivation and intuition for the method, or perhaps do some some analysis could be done to indicate why it's working (where it is).
- The paper motivates DUEL as learning to represent input sequences in a way that facilities compositional generalization, but it's not totally clear to me how the alternating freezing does this. It seems like the meta-learning approach (which directly trains for compositional generalization) that other work has employed is more directly suited to this, or even perhaps some adversarial training approach if the paper is aiming to learn representations f(x) that encode invariances across s and ~s.
- It would help if the paper could somehow characterize the representations or models that DUEL learns (e.g. providing something like a fixed-point analysis), which "the algorithm has converged to the desired representation since the difference in representing s and ~s is small" in section 4 starts to do, but wasn't totally clear to me.
- Or alternatively, perhaps the paper could do some empirical investigation of distributional differences in f(x) when x is drawn from s versus from ~s.

W3) Some of the choices in the design of the method felt a bit arbitrary, and if they were better justified I'd feel more confident that the approach is working for understandable reasons.
- Why reinitialize the parameters of g in fine-tuning?
- The pre-finetuning setup on p does not match the fine-tuning setup on q in that fine-tuning updates both f and g. Why not update f in training on p, or keep f fixed in training on q?
- While in some compositional generalization tasks, ~s is intuitively harder (e.g. longer inputs and outputs) than s in some way, in other tasks s and ~s are interchangeable, so does it matter that g is updated on s and f on ~s (and that the last updates in pre-finetuning are always done on ~s)?

W4) It would help to present statistical significance results, or standard deviations across multiple seeds, as it's a bit difficult to interpret the significance of the improvements. However, since the improvements are consistent (albeit small), I don't think this is a crucial weakness.

*Minor comments*

- It would help to give some intuition for the \alpha in Section 3. How is this value chosen?
- The update equations (1-3) with a single step-size make it seem like SGD is being used, but from the appendix it's Adam.
- How are the logical forms updated in the COGS_VAR splits to match the changes to the input sentences?

*Typos*:

- pg 4 "standard supervise learning" -> "standard supervised learning"
- pg 5: several grammatical errors at the end of section 4
- pg 6: "BERT_SMALL" -> "BERT_BASE" (?)
- pg 6: "GEO_TMCD2" -> GEO_{cd}"
- pg 7: "When Will DUEL Works Best" -> "When Will DUEL Work Best"
- pg 7: "compsitional" -> "compositional"
- pg 7: "DUEL helps extracting" -> "DUEL helps extract"


**Summary Of The Paper:**

This paper proposes a training procedure for encoder--decoder models (applied to semantic parsing) which aims to improve the models' ability to compositionally generalize (successfully handle novel combinations of words and structures, where combinations were not seen in training). The approach relies on pre-finetuning: training a model on a different dataset than the target dataset that also the requires the same sort of compositional generalization as the target dataset, before then training on the training set of the target dataset and then evaluating zero-shot on the compositional set of the target dataset, in the standard way. In pre-finetuning, the decoder is only updated on the training set while the encoder is updated on the compositional generalization set. The approach is evaluated using two different pre-trained encoder--decoder transformer architectures on three different semantic parsing compositional generalization datasets from past work, where it obtains consistent (albiet somewhat small) improvements over a baseline that pre-finetunes all model parameters, and outperforms a past state-of-the-art model on one dataset.

**Summary Of The Review:**

I feel a bit borderline about this paper, as the method seems a bit limited and heuristic -- not being clearly designed for compositional generalization or showing convincing results on the hardest tests of compositional generalization. But, it does seem to show consistent (if sometimes small) improvements on a couple models and several datasets, the methodology seems sound, and the paper is very clearly written. I've put an overall score of 5 for now, but I look forward to discussion.

---

Update after the response: Thanks to the authors for their thorough response to my comments! The explanations and new ablation results helped convince me that the choices made in designing the method were reasonable. I also appreciate the standard deviations, which make me confident that the improvements are real. I'm in favor of accepting this paper, and have updated my score to a 6 (from a 5).

---

> ### Author Response · Authors · 2021-11-22
> **Response to Reviewer K5oD**
>
> **Comment 1:** *“I wasn't totally convinced that the method works well on strong tests of compositional generalization.”*
>
> **Response:** The limited performance of our method on COGS structural generalization could be related to a specific pathology of neural models, which struggle to extrapolate to longer output lengths than those seen during training [1]. Note that the COGS training set has an average input length of 7.48, its structural generalization set has an average input length of 24.04 (due to the examples’ nested recursion). Our method, in its current form, does not seem to fully address this problem.  The example of COGS_VAR further highlights this challenge and the need for future work.
>
> **Comment 2:** *“It's not totally clear to me why the method should enable compositional generalization in general, and I feel like it would help to strengthen the motivation and intuition for the method, or perhaps some analysis could be done to indicate why it's working (where it is).”*
>
> **Response:** We agree with you (and other reviewers). We have been working in this direction. Please see our response to the Reviewer uYpx’s Comment 2 and 3.
>
> **Comment 3:** *“Some of the choices in the design of the method felt a bit arbitrary, and if they were better justified I'd feel more confident that the approach is working for understandable reasons.”*
>
> **Response:**
> * *"reinitialize the parameters of g"*: The intuition is to eliminate the decoder’s task-specific inductive bias from the pre-finetuning task and to only keep the pre-finetuned encoder with the learned compositional ability.
> * *"Why not update f in training on s"*: f is for learning compositional abilities, training on s will make it overfit to solving specific tasks.
> “Why not keep f fixed in fine-tuning on q?”: We tried to keep f fixed in fine-tuning, results are worse than our method. A brief summary of the results is given below (in the row of DUEL(F)):
>
> **Accuracy on the target tasks after pre-finetuning on other tasks (splits in the CG General category)** \
> *| Model &nbsp;&nbsp;&nbsp;&nbsp;&nbsp;&nbsp; | Method &nbsp;&nbsp;  | &nbsp;&nbsp; GEO &nbsp;&nbsp; | &nbsp;&nbsp; SCAN &nbsp;&nbsp; | COGS | SCAN | COGS | GEO |* \
> *| &nbsp;&nbsp;&nbsp;&nbsp;&nbsp;&nbsp;&nbsp;&nbsp;&nbsp;&nbsp;&nbsp;&nbsp;&nbsp;&nbsp;&nbsp;&nbsp;&nbsp;&nbsp; | &nbsp; &nbsp;&nbsp;&nbsp;&nbsp;&nbsp;&nbsp;&nbsp;&nbsp;&nbsp;&nbsp;&nbsp;&nbsp;&nbsp;  | &nbsp;&nbsp;&nbsp;&nbsp;&nbsp; → COGS_cg &nbsp;&nbsp;&nbsp;&nbsp;&nbsp;&nbsp; | &nbsp;&nbsp;   → GEO_cd&nbsp;&nbsp; | → SCAN_cd &nbsp;   |* \
> | BERT2SEQ | DUEL(F)  |  61.4/1.0
>  | 58.6/1.1 | &nbsp; 35.8  | 36.1 &nbsp; | &nbsp; 4.4 &nbsp; | &nbsp;4.8&nbsp; | \
> | BERT2SEQ | DUEL   &nbsp;&nbsp;&nbsp;&nbsp;   | 75.5/1.1 | 73.2/1.1 | &nbsp; 36.9  | 37.7 &nbsp; | &nbsp; 4.8 &nbsp; | &nbsp;5.5&nbsp; | \
> | T5-Base  &nbsp;&nbsp;&nbsp;   | DUEL(F)  | 90.7/2.5 | 89.6/2.5 | &nbsp;  52.5 | 55.7 &nbsp;  | &nbsp; 16.0 | 18.9 | \
> | T5-Base  &nbsp;&nbsp;&nbsp;   | DUEL  &nbsp;&nbsp;&nbsp;&nbsp;     | 95.4/4.5 | 94.9/3.9 | &nbsp; 53.8  | 57.8 &nbsp; | &nbsp; 20.2 | 21.3 |
>
> * *“does it matter that g is updated on s and f on ~s ”:*  Yes, it does matter for tasks with asymmetric splits (such as GeoQuery’s TMCD splits). Please see our response to Reviewer gjvP’s Comment 1.
>
> **Comment 4:** *“It would help to present statistical significance results, or standard deviations across multiple seeds, as it's a bit difficult to interpret the significance of the improvements.”*
>
> **Response:** We would like to take this opportunity to clarify that the results in all the tables are averaged over 3 runs. We add standard deviations for DUEL of Table 2 and Table 3 in the Appendix D.4 of the revised manuscript.
>
> **Comment 5:** *“Minor comments.”*
>
> **Response:**
> * *“intuition for the \alpha”*: We use the same \alpha as in Shaw et.al. [2].
> * *“equations (1-3) seem like SGD but Adam in Appendix”*: We use equations (1-3) to illustrate the computation of gradient. We will clarify in the revised manuscript.
> * *“logical forms updated in the COGS_VAR”*: Same as the input utterances, the logical forms are replaced by the word replacement mapping. We add output programs in Appendix C of the revised manuscript.
>
>
> [1] Benjamin Newman, John Hewitt, Percy Liang, Christopher D. Manning. The EOS Decision and Length Extrapolation. Proceedings of the Third BlackboxNLP Workshop on Analyzing and Interpreting Neural Networks for NLP, pp. 276-291, 2020\
> [2] Peter Shaw, Ming-Wei Chang, Panupong Pasupat, Kristina Toutanova. Compositional Generalization and Natural Language Variation: Can a Semantic Parsing Approach Handle Both? Proceedings of the 59th Annual Meeting of the Association for Computational Linguistics and the 11th International Joint Conference on Natural Language Processing, pp. 922–938, 2021

---

### Official Review · Reviewer_gjvP · 2021-11-03

**Correctness:** 3
**Technical Novelty And Significance:** 3
**Empirical Novelty And Significance:** 3
**Recommendation:** 6
**Confidence:** 4

**Main Review:**

Overall I thought that this was an interesting paper, which was mostly clearly written and organized, and generally I liked the method that was introduced. I'm leaning toward acceptance (and I can imagine the concerns below being addressed satisfactorily and further increasing my confidence).

I had a few questions and potential concerns that weaken my confidence in the impact of the contribution.

The first question involves the reasoning behind the particular design of the method. The authors lay out a rationale for training the encoder parameters on the test component of each split, and the decoder on the train component of each split -- but the reasoning given is not terribly transparent to me, and I was left wondering whether similar results could be achieved by instead training the encoder on the train component and the decoder on the test component. Was this something that the authors tried? I think that including this comparison could be informative with respect to the importance of setting up the method in this specific way.

Another confusion I had involved the original purpose of these various datasets, and how this related to the current usage. For instance, not being familiar with GeoQuery, the description in the paper led me to believe that it was designed as a QA dataset, so I was wondering why any SOTA would exist for semantic parsing on a QA dataset. A google search suggests that GeoQuery is in fact annotated with semantic parses, but this confusion could be alleviated by making clearer that the dataset is used for semantic parsing.

I was similarly wondering about use of SCAN for semantic parsing, since my understanding was that SCAN was designed for mapping commands to actions. If this is correct, where are the semantic parses coming from for that dataset? I imagine that since it is synthetic/template-based, producing semantic parses in a rule-based manner may be straightforward, but it wasn't clear to me from the paper how this was working.

My additional concerns are focused more on the impact of the contribution. The baselines that the paper compares against are for the most part not external models -- rather, the authors are comparing only against baseline versions of their own model, without the key components of the new method. So the improvement over the baselines indicates that the method does improve over the same model without the iterative compositional-split training. However, it is only in the one case of the GeoQuery dataset that the authors mention the existing SOTA (which they have beaten), suggesting that there are stronger SOTA models on the other datasets (or at least on COGS, if SCAN is not used typically for semantic parsing?). What this leads me to believe is that while the method improves over a vanilla model, it may not be improving over stronger models that use alternative methods for COGS (and possibly SCAN). I'm curious especially whether other models have made better headway on the COGS structural test, which is showing especially low performance here. It would be helpful to get greater clarity on how the presented results relate to the strongest existing results from other models across all datasets.

Finally, I'm not totally sure how surprised/impressed we should be by improvements from this method. Specifically, I'm wondering how impressed we should be that we see a performance boost from training models to generalize across a specific type of split (e.g., one in which the test sentences are longer than the training sentences), for exactly the target task (semantic parsing). The authors make this general observation in Section 5.3, when they acknowledge that the model works best when the compositional splits are maximally similar. So to what extent is it ultimately somewhat obvious that training models to handle a given type of split will help it on this type of split?

To put the above concern another way: to what extent are we potentially *no longer testing models on compositional generalization* if we train them directly to be able to generalize in the particular way that is needed for the selected compositional split? If the performance boost is very specific to a particular relationship between test and training data, is this simply allowing the model to learn strategies specific to that particular type of generalization, such that it no longer needs to use composition to achieve that generalization? This would defeat the purpose of trying to improve models' ability to use actual compositional processing to show the desired generalization. So I would like to see the authors address this concern.


**Summary Of The Paper:**

This paper is focused on the problem of compositional generalization in semantic parsing, and introduces a method called "DUEL", which involves "pre-finetuning" iteratively on compositional train-test splits from other datasets, before transferring to fine-tuning on the training data from the target dataset. The method involves using the compositional train/test split from one dataset, and training their encoder-decoder model iteratively such that the encoder parameters are updated based on the test data from that dataset, and the decoder parameters are updated based on the training data from that dataset. After this "pre-finetuning", the model is fine-tuned on the training data from the target dataset. They find that their model outperforms baselines involving 1) fine-tuning on the target task only, and 2) pre-finetuning on the merged data from the other dataset, without the encoder/decoder split. They find that their method largely does not help with the extremely low numbers on COGS structural items, but the margins of improvement are larger for GeoQuery data and SCAN data, with the authors claiming a new SOTA result on one of the splits for GeoQuery.

**Summary Of The Review:**

In general I found the method interesting and the paper overall clear. However, I have some remaining questions about certain aspects of the method and datasets/tasks, as well as some potential concerns about the impact of the contribution, which I would like to see addressed before I can strongly endorse this paper.

---

> ### Author Response · Authors · 2021-11-22
> **Response to Reviewer gjvP**
>
> **Comment 1:** *“whether similar results could be achieved by instead training the encoder on the train component and the decoder on the test component.”*
>
> **Response**: We followed the suggestion and provided the comparison - the new results are as follows, in the row of DUEL(S):
>
> **Accuracy on the target tasks after pre-finetuning on other tasks (splits in the CG General category)** \
> *| Model &nbsp;&nbsp;&nbsp;&nbsp;&nbsp;&nbsp; | Method &nbsp;&nbsp;  | &nbsp;&nbsp; GEO &nbsp;&nbsp; | &nbsp;&nbsp; SCAN &nbsp;&nbsp; | COGS | SCAN | COGS | GEO |* \
> *| &nbsp;&nbsp;&nbsp;&nbsp;&nbsp;&nbsp;&nbsp;&nbsp;&nbsp;&nbsp;&nbsp;&nbsp;&nbsp;&nbsp;&nbsp;&nbsp;&nbsp;&nbsp; | &nbsp; &nbsp;&nbsp;&nbsp;&nbsp;&nbsp;&nbsp;&nbsp;&nbsp;&nbsp;&nbsp;&nbsp;&nbsp;&nbsp;  | &nbsp;&nbsp;&nbsp;&nbsp;&nbsp; → COGS_cg &nbsp;&nbsp;&nbsp;&nbsp;&nbsp;&nbsp; | &nbsp;&nbsp;   → GEO_cd&nbsp;&nbsp; | → SCAN_cd &nbsp;   |* \
> | BERT2SEQ | DUEL(S)  | 72.0/0.3 | 72.3/1.3 | &nbsp; 36.3  | 37.5 &nbsp; | &nbsp; 5.1 &nbsp; | &nbsp;2.0&nbsp; | \
> | BERT2SEQ | DUEL   &nbsp;&nbsp;&nbsp;&nbsp;   | 75.5/1.1 | 73.2/1.1 | &nbsp; 36.9  | 37.7 &nbsp; | &nbsp; 4.8 &nbsp; | &nbsp;5.5&nbsp; | \
> | T5-Base  &nbsp;&nbsp;&nbsp;   | DUEL(S)  | 94.8/2.3 | 95.1/3.6 | &nbsp;  53.4 | 57.5 &nbsp;  | &nbsp; 20.1 | 12.7 | \
> | T5-Base  &nbsp;&nbsp;&nbsp;   | DUEL  &nbsp;&nbsp;&nbsp;&nbsp;     | 95.4/4.5 | 94.9/3.9 | &nbsp; 53.8  | 57.8 &nbsp; | &nbsp; 20.2 | 21.3 |
>
> DUEL(S) shows minor degradation to DUEL when pre-finetuning on SCAN, likely due to its symmetric MCD split. Similarly, the COGS lexical generalization set has almost the same average input length as the train set and contains all the vocabulary. Thus, COGS training set is largely symmetric with its lexical generalization set. As we have analyzed in Sec 5.4 that DUEL does not learn much from the structural generalization, a minor change in DUEL(S) is reasonable. However, DUEL(S) has a large gap with DUEL when pre-finetuning on GeoQuery. As the TMCD split only ensures all atoms appear in the TMCD1 component, we find it has ~30% of examples containing words that do not appear in the test component (ie, TMCD2). Thus, using TMCD2 to train the decoder and TMCD1 to train the encoder forces the encoder to model those “new” atoms. This likely causes the encoder not learning enough inductive bias, thus incurring a larger drop in performance.
>
> We add the results in the Appendix D.3 of the revised manuscript.
>
> **Comment 2:** *“The original purpose of these various datasets, and how this related to the current usage”*
>
> **Response:** We will clarify the description of GeoQuery, which is indeed a semantic parsing dataset and a widely used benchmark for that task. We will also clarify the description of SCAN, which maps natural language commands to action sequences, and is commonly used as a diagnostic dataset for evaluating compositional generalization. We add the clarification in the revised manuscript.
>
> **Comment 3:** *“The baselines that the paper compares against are for the most part not external models... how the presented results relate to the strongest existing results from other models across all datasets.”*
>
> **Response:** Please see our response to Reviewer uYpx’s Comment 1.
>
> **Comment 4:** *“To what extent are we potentially no longer testing models on compositional generalization if we train them directly to be able to generalize in the particular way that is needed for the selected compositional split?”*
>
> **Response:** We agree that a method for compositional generalization is less useful if it only improves performance in the context of a very specific type of distribution shift. To extend your point about learning “general” skills for compositional generalization, please see our responses to Reviewer uYpx’s comments 2 and 3.
>
> In short, we believe it is possible to gain such generic skills by pre-training or pre-finetuning many tasks that are in compositional generalization style.  Our results are indeed two folded: first, we show the performance on the target task could be improved even if we pre-finetune on a task that is not obviously related. Secondly, we do show a specific type aspect of composition generalization, ie, length split, could be improved more pronouncedly. Thus, the future direct would be to construct many different types of compositional generalization tasks for pre-training or pre-finetuning.

---

### Official Review · Reviewer_uYpx · 2021-11-05

**Correctness:** 3
**Technical Novelty And Significance:** 3
**Empirical Novelty And Significance:** 2
**Recommendation:** 5
**Confidence:** 4

**Main Review:**

While a like this idea, there are several issues with the proposal approach:

1. **Comparison with State-of-the-Art** There is little information in Section 5 about comparison with existing approaches in compositional generalization for semantic parsing. Indeed, in recent years several seminal works have emerged, pushing accuracies on some synthetic tasks like SCAN to near 100% accuracy. These works are not mentioned in Section 5. While the model outperformed the currently best  approach (Shaw et al., 2021) on GEO_{TMCD}, the lower results on other simpler tasks like SCAN make me feel a bit concerned about the results. Perhaps this is because only few models are evaluated on GEO_{TMCD} so far, and previous approaches more tailored to the context-free utterances on GEO (e.g., Herzig and Berant) would actually perform better?

2. **Methodology** Another issue is related to the proposed approach itself. While pre-finetuning encoder on compositional split A and the decoder on compositional split B could encourage the encoder to learn representations that generalize better to split B, the generalization strategy learned by the encoder might be specific to split B only, and might not be able to generalize to other compositionally novel distributions (e.g., the final evaluation data). Ideally during the pre-finetuning stage the model need to learn to generalize well to *arbitrary* mismatched splits, but only presenting one set of splits (A/B) might not be enough for the model to learn a more "general-purpose" strategy.

3. **Transferability across Datasets** Transfer learning on NLP tasks would require the source and target domains share reasonably amount of common language patterns in order to perform well. However, the tasks used in this paper have drastically different utterances in language styles and compositional patterns, which makes transfer learning quite non-trivial. For example, SCAN only contains toyish words like JUMP and simple composition strategies like concatenation (JUMP TWICE). It would be doubtable if learning to generalize well on this toyish domain would be useful for handling real-world utterances with diverse language style, like GEO. The authors could present more analysis in terms of the language and compositionality styles of those datasets in order to have a better understanding of the upper-bound performance of transfer learning approaches for compositional generalization.

**Summary Of The Paper:**

This paper presents a transfer learning strategy for improving compositional generalization of semantic parsers based on pre-trained language models. Before fine-tuning the model on data from the target domain, the authors propose a pre-finetuning step, where models are trained on compositional splits of data from another source domain, with the goal to transfer the model's learned knowledge about language compositionality during this pre-finetuning step to the final learning stage on the target domain, therefore improving compositional generalization. To this end, the authors propose a pre-finetuning method which encourages the model to discover representations of natural language that are invariant against its compositional structures. This is achieved by iteratively freezing the encoder or decoder modules during pre-finetuning, and training the encoder and decoder modules on compositionally disjoint splits of the source data, such that the encoder learns representations that are robust against distributional shift of language compositionality.



**Summary Of The Review:**

This paper presents a nice idea for improving compositional generalization of neural semantic parsers. The results on GEO_{TMCD} outperforms the currently best approach. However, there are issues with both experimentation and the methodology.

---

> ### Author Response · Authors · 2021-11-22
> **Response to Reviewer uYpx (1/2)**
>
> **Comment 1:**  *"Comparison with Other Methods"*
>
> **Response:**  In this work, we focus on developing general learning recipes that are agnostic about the specific designs of the datasets. Our main consideration is that several existing works have already shown that  solving SCAN with specially designed approaches do not transfer to non-synthetic tasks, see [1, 2] for a review. Others such as SpanBasedSP (SBSP) and NQG-T5 are more successful.  Thus, following your suggestion, we compare to these two as well as a recently proposed meta-learning approach [3]. The results are reported in Appendix D.1 in the revised manuscript and presented also in below:
>
> Method       | COGS_cg | GEO_cd | SCAN_cd
> :---|:---:|:---:|:---:
> SBSP         |     -     |     49.2    |     100
> NQG-T5     |    -    |     56.6    |     100
> MAML     |  92.7/3.1    |    -    |     15.9
> DUEL     |  95.4/4.5    |     57.8    |     21.3
>
> DUEL outperforms MAML on SCAN_CD and COGS, and outperforms SBSP substantially  on GEO_CD.  Despite SCAN's usefulness in highlighting certain aspects of compositional generalization, we do not believe a new method has to solve it perfectly in order to be scientifically meritorious.
>
> **Comment 2:** *“..one set of splits (A/B) might not be enough for the model..."general-purpose"”*
>
> **Response:** Your suggestion is interesting. To clarify, our interest is to investigate whether *any* pre-finetuning task with a compositional generalization split could benefit the target task.
> Here we report a related experimental result in a similar vein to what you had suggested. In the revised Table 3,  we show we can improve the performance on the target task Geo_len by pre-finetuning on *two* different tasks: COGS_cg and SCAN_len than by pre-finetuning just on one of them:
>
> Model    |      Method    |  COGS_cg | SCAN_len | COGS_cg + SCAN_len
> :---|:---:|:---:|:---:|:---:
> BERT2SEQ    |   MERGED    |      16.8    |    16.1    |     17.0
> BERT2SEQ    |       DUEL    |      17.7    |    18.6    |     18.9
> T5-Base    |   MERGED    |      40.9    |    39.1    |     41.4
> T5-Base    |       DUEL    |      43.0    |    45.0    |     46.4
>
> To test your hypothesis specifically, one has to come up with multiple *different* CG splits for a pre-finetuning task. So far, the most common ones are length split, splits that test specific elements (such as non-overlapping primitives etc) and the more generalized (T)MCD splits. We can generate many random (instantiation of)  such splits types but we feel the proposed approach would be particularly useful if we could have many categorically different *types* of splits.  We leave this to future work.

---

> ### Author Response · Authors · 2021-11-22
> **Response to Reviewer uYpx (2/2)**
>
> **Comment 3:** *“more analysis.. better understanding…  of transfer learning approaches for compositional generalization”*
>
> **Response:** We completely agree. Our results on the transferability from different pairs of pre-finetuning and target tasks, without obvious common language patterns, are intriguing. (We thoroughly examined our experimental protocols and found no error.)
>
> We have been testing a few hypotheses. One theory is that the DUEL pre-finetuning regularizes the encoder to represent structural information in a more decomposable way, instead of “contextually interdependent”.  Suppose the input sentence is “jump right after walk right”.  Ideally, we would want the encoder to pick up the cue that “after” signifies to represent “jump right” and “walk right” separately. However, for an end-to-end model, the representation could be biased to couple these two shorter phrases together to map to the desired output sequence.
>
> This theory is appealing as it does not directly appeal to common language patterns such as those studied and observed in synchronous context-free grammars and span alignments (e.g. cite Herzig et al. 2021, Shaw et al. 2021, Liu et al. 2020, Chen et al. 2020). Rather, this theory hypothesizes that the pre-training tasks for pre-training language models do not have enough inductive biases from compositional generalization types of tasks to learn decomposable representations.
>
> We have not come up with a definitive way to test the theory -- a clean way would be pre-training language models with (many) large-scale compositional generalization tasks. Our pre-finetuning tasks inject the desired inductive biases after the standard pre-training.
>
> We added several indirect evidences to support the theory, in Appendix D.2. We compare the input length of correctly predicted examples in Geo_len between training with DUEL and MERGED as follows:
>
> Model  | Method |  SCAN_std | SCAN_cd | COGS_cg | SCAN_len
> :---|:---:|:---:|:---:|:---:|:---:
> BERT2SEQ | None | 7.633 (71)
> BERT2SEQ | MERGED | 7.634 (71) | 7.634 (71) | 7.642 (74) | 7.634 (71)
> BERT2SEQ | DUEL       | 7.620 (70) | 7.651 (74) | 7.657 (78) | 7.693 (81)
> T5-Base | None | 8.876 (170)
> T5-Base | MERGED | 8.872 (172) | 8.872 (172) | 8.899 (180) | 8.872 (172)
> T5-Base | DUEL       | 8.865 (173) | 8.960 (176) | 8.968 (189) | 8.985 (198)
>
> We observe that pre-finetuning improves on longer inputs in both the average length and the number of them.
>
>
> [1] Daniel Furrer, Marc van Zee, Nathan Scales, Nathanael Schärli. Compositional Generalization in Semantic Parsing: Pre-training vs. Specialized Architectures. Arxiv 2021 \
> [2] Peter Shaw, Ming-Wei Chang, Panupong Pasupat, Kristina Toutanova. Compositional Generalization and Natural Language Variation: Can a Semantic Parsing Approach Handle Both? Proceedings of the 59th Annual Meeting of the Association for Computational Linguistics and the 11th International Joint Conference on Natural Language Processing, pp. 922–938, 2021 \
> [3] Henry Conklin, Bailin Wang, Kenny Smith, and Ivan Titov. Meta-learning to compositionally generalize. Proceedings of the 59th Annual Meeting of the Association for Computational Linguistics and the 11th International Joint Conference on Natural Language Processing, pp. 3322–3335, 2021.

---

### Author Response · Authors · 2021-11-22
**General Response**

We thank all the reviewers for their efforts and comments.  We have submitted a revised manuscript. In particular,

* We provide additional results to compare to other methods, as well as ablation studies as suggested in the reviews.

* We motivate and provide a stronger rationale/hypothesis for the proposed method.  We do concur with all the reviewers that it is intriguing that the proposed method can improve the performance on the target tasks which have no obvious common language patterns with pre-finetuning tasks. Our intuition is that the encoder is regularized through the pre-finetuning to learn more decomposable representation of linguistic constituents.

Revisions and new materials (results and texts) are marked in blue. For convenience, most new materials are added to the appendix sections.

---

### Decision · Program_Chairs · 2022-01-20

**Decision:**

Reject

**Comment:**

The authors attempt to tackle the problem of compositional generalization, i.e., the problem of generalizing to
novel combinations of familiar words or structures. The authors propose a transfer learning strategy based on
pretraining language models. The idea is to introduce a pre-finetuning task where a model is first trained on compositional train-test splits from other datasets, before transferring to fine-tuning on the training data from the target dataset. Although the technique
brings some improvements, and the authors do their best the address the reviewers' questions, it is still unclear:

a) Why the method should work in principle, whether there is a theoretical backing and how it formally relates to meta-learning
b) How the approach compares to data augmentation methods since pre-finetuning requires more data, albeit from a different
dataset. See for example: https://openreview.net/forum?id=PS3IMnScugk
c) The whole approach would be more convincing if the authors could articulate *how* their method renders a model
more robust to distribution shifts (e.g., based on GOGS results it does not help structural generalization, do the gains
come from lexical generalization?)
d) it would also be interesting whether this method works on larger scale or more realistic datsets like CFQ, ATIS or machine translation
https://arxiv.org/pdf/1912.09713.pdf
https://arxiv.org/abs/2010.11818